

# Precessional pacing of tropical ocean carbon export during the Late Cretaceous

Ji-Eun Kim[1], Thomas Westerhold[2], Laia Alegret[3], Anna Joy Drury[4], Ursula Röhl[2], Elizabeth M. Griffith[1]

[1]School of Earth Sciences, The Ohio State University, 43210 USA
[2]Center for Marine Environmental Sciences (MARUM), University of Bremen, 28359 Germany
[3]Department of Earth Sciences, University of Zaragoza, 50009 Spain
[4]Department of Earth Sciences, University College London, WC1E 6BT UK

*Correspondence to*: Elizabeth M. Griffith (griffith.906@osu.edu) and Thomas Westerhold (twesterhold@marum.de)

**Abstract.** The marine biological carbon pump, which exports organic carbon out of the surface ocean, plays an essential role
in sequestering carbon from the atmosphere, thus impacting climate and affecting marine ecosystems. Orbital variations in
solar insolation modulate these processes, but their influence on the tropical Pacific during the Late Cretaceous is unknown.
Here we present a high-resolution composite record of elemental barium from deep sea sediments as a proxy for organic carbon
export out of the surface oceans (i.e., export production) from Shatsky Rise in the tropical Pacific. Variations in export
production in the Pacific during the Maastrichtian, from 71.5 to 66 million years ago, were dominated by precession and less
so by eccentricity modulation or obliquity, confirming that tropical surface-ocean carbon dynamics were influenced by
seasonal insolation in the tropics during this greenhouse period. We suggest that precession paced primary production in the
tropical Pacific and recycling in the euphotic zone by changing water column stratification, upwelling intensity, and continental
nutrient fluxes. Benthic foraminiferal accumulation rates covaried with export production providing evidence for bentho-
pelagic coupling of the marine biological carbon pump across these high-frequency changes in a cool greenhouse planet.

**Short Summary**

This study attempts to get a better understanding of the marine biological carbon pump and ecosystem functioning under
warmer than today conditions. Our records from marine sediments show the Pacific tropical marine biological carbon pump
was driven by variations in seasonal insolation in the tropics during the late Cretaceous and may play a key role in modulating
climate and the carbon cycle globally in the future.

**1 Introduction**

High-resolution marine sedimentary records are important for shedding new light on critical aspects of Earth's climate and
feedbacks in the ocean-atmosphere-biosphere system. On time scales of 10 thousand to 1 million years, variations in Earth's
climate are paced by orbital (Milankovitch) cycles of eccentricity, obliquity and precession (e.g., 100 kyr, 41 kyr, and 23-18



kyr) because they alter solar insolation and its distribution on Earth (Laskar et al., 2004, 2011). Orbital-scale climate change

is well studied for the Cenozoic Era (e.g., Zachos et al., 2001; Westerhold et al., 2020), but there is limited understanding for earlier times prior to 66 million years. After peak hot greenhouse conditions of the Cretaceous during the Cenomanian about 100 million years ago (e.g., Forster et al., 2007; Huber et al., 2018), progressive global cooling ensued and continued until the end of the Cretaceous (Jenkyns et al., 1994; Friedrich et al., 2012; Voigt et al., 2012). Cool and warm periods were overlaid on the long-term climatic cooling, e.g., the early Maastrichtian cooling pulse (Haynes et al., 2020) and the warm mid-

Maastrichtian event (Frank et al., 2005). Extinction of the globally distributed inoceramid bivalves also occurred following a peak in their abundance (e.g., MacLeod and Huber, 1996; Dameron et al., 2017). However, our understanding of the relationship between environmental and biotic change during this time is limited, in part due to the lack of records of key drivers of variability in the ocean-carbon-climate system at orbital timescales in key regions like the tropical Pacific.

Many lines of evidence including climate modeling studies suggest that low- and mid-latitude processes in the climate

system respond in nonlinear ways to insolation forcing (e.g., Short et al., 1991; Crowley et al., 1992; Laepple and Lohmann, 2009; Zeebe et al., 2017). In the low-latitudes, eccentricity-modulated precession is expected to dominate carbon cycle dynamics (Herbert et al., 1999; MacLeod et al., 2001). A key feedback driving these dynamics seems to relate to the global redistribution of moisture and energy by the hydrological cycle (e.g., Trenberth et al., 2000). Proxy records that provide evidence of carbon cycle dynamics, such as marine carbon isotope records ($\delta^{13}C$), show dominant variability in the eccentricity

(rather than precession) band. This effect could be due to the long residence time of carbon in Earth's exogenic system, which filters out higher resolution fluctuations (e.g., Pälike et al., 2006) or is related to orbitally paced phytoplankton evolution (Beaufort et al., 2022). Nonetheless, during the Late Cretaceous several records outside the tropical Pacific show precession-scale changes in sedimentology and bulk carbonate and foraminiferal carbon isotopes (Herbert et al., 1999; MacLeod et al., 2001; Batenburg et al., 2012, 2014; Barnet et al., 2018, 2019). Whether or not the tropical Pacific carbon cycle, which plays

an important role in climate regulation as the largest ocean basin with high burial of pelagic biogenic sediment due to equatorial upwelling, was paced by precession is not clear. The resolution of existing records is insufficient to resolve precessional cycles (Jung et al., 2012, 2013; Voigt et al., 2012; Dameron et al., 2017). Without sufficient sampling resolution of existing records, high-frequency variability in the Earth system could potentially bias conclusions based on unrepresentative or less resolved records (e.g., Herbert et al., 1999).

Here we present an astronomically-tuned, high-resolution record of open ocean organic carbon export, the organic carbon flux out of the surface of the ocean or export production, in the tropical Pacific during the Maastrichtian from 71.5 to 66 million years ago (Fig. 1). Export production is a key organic carbon flux within the marine biological carbon pump which sequesters carbon from atmosphere to the deep ocean through biologically mediated processes. We use a new composite X-ray fluorescence (XRF) scanning barium (Ba) record as a proxy for export production by comparing it to pelagic marine barite

extracted from discrete samples and excess- or bio-Ba measurements on the same samples from the same cores. Accumulation of pelagic marine barite and excess-Ba in deep-sea sediments have been shown to correlate with organic carbon flux out of the photic zone (Paytan et al., 1996; Eagle et al., 2003), making these good proxies for export production where there is low





terrigenous input and sediment is not sulfate-reducing. We also reconstruct benthic foraminiferal assemblages and accumulation rates over two windows of time to test the relationship between organic carbon export from the surface ocean

represented by the Ba records and organic carbon reaching the seafloor (i.e., the sequestration flux) represented by the benthic foraminiferal records. These new records allow us to better characterize the marine biological pump at this time and show that precession, rather than eccentricity or obliquity, was the dominant orbital cycle that paced ocean export production thereby influencing the marine carbon cycle in the tropics during the Maastrichtian cool greenhouse period.

## 2 Materials and methods

### 70 2.1 Geological setting

Shatsky Rise formed about 145 million years ago in the Pacific (Larson et al., 1992; Sager et al., 1999), providing an archive of Cretaceous sediments thought to have been above the carbonate compensation depth (CCD) between 1500 to 2000 m throughout their deposition (Sager et al., 1999). The current location of ODP Leg 198 Sites 1209 and 1210 are 32°39.1081′N, 158°30.3564′E (2387 m below sea level) and 32°13.420′N, 158°15.5623′E (2573 m below sea level), respectively (Bralower

et al., 2002). Shatsky Rise drifted northward on the Pacific plate to its present location (Larson et al., 1992; Dameron et al., 2017). During the Late Cretaceous, Shatsky Rise was approximately 10ºN of the equator (Fig. 1; Larson et al., 1992; Dameron et al., 2017) in a region of relatively high productivity due to upwelling cause by the divergence of the North Equatorial Counter Current and North Equatorial Current.

### 2.2 Samples

This study used deep-sea marine sediment samples from Ocean Drilling Program (ODP) Leg 198 Sites 1209 and 1210 on Shatsky Rise, Northwest Pacific during the Maastrichtian (from 71.5 to 66 million years ago). These samples consisted primarily of very pale orange to white nannofossil ooze, mostly > 96 weight percent (wt %) $CaCO_3$, and discontinuous chert nodules likely formed by remobilized biogenic silica (Bralower et al., 2002). Bulk carbonate $\delta^{18}O$ and $\delta^{13}C$ were measured in 1394 samples from both Sites 1209 (1096) and 1210 (298) about every 5 cm. Discrete samples from 5 intervals that consist of

51 samples in total included Samples 1209C-16H-1, 114 to 145 cm, -21H-1, 67 to 118 cm, -22H-2, 133 to -3, 31 cm, Sample 1209B-29H-1, 87 cm to -2, 20 cm, and Sample 1210B-25H-4, 43 to 80 cm. Samples were taken every 5 cm or so in the core and weigh between 13 to 22 g (approximately 20 cm³). Additional samples for benthic foraminifera were taken from over 2 of the 5 intervals which have new barite data.

### 2.3 Bulk stable carbon and oxygen isotopes

Bulk carbonate $\delta^{13}C$ and $\delta^{18}O$ analyses (Datasets S1-S3) on 1394 freeze-dried and pulverized sediment samples from Sites 1209 (1096) and 1210 (298) were performed at MARUM, University Bremen. The bulk stable carbon and oxygen isotope data are reported relative to the Vienna Pee Dee Belemnite (VPDB) international standard, determined via adjustment to calibrated





in-house standards and NBS-19. Bulk carbonate analyses were carried out on a Finnigan MAT 251 mass spectrometer equipped with automated carbonate preparation line (Kiel III). Carbonate was reacted with orthophosphoric acid at 75 °C. Analytical
precision based on replicate analyses of in-house standard (Solnhofen Limestone) is 0.04‰ (1σ) for δ¹³C and 0.07‰ (1σ) for δ¹⁸O.

## 2.4 Age Model

An initial 405 kyr age model was first applied using the shipboard biostratigraphic datums (Fig. S1) and the Shatsky Rise ODP Sites 1209 and 1210 composite bulk δ¹³C filtered for the 405 kyr cycles in the depth domain (5 m bandpass filter using 0.2 ±
0.06). Then the thirteen identified 405 kyr cycles were correlated to the Laskar et al. (2004) and Laskar (2020) cosine function with correlating maxima in the δ¹³C filter to minima in the cosine function (Fig. S2). This phase relationship is following the findings in the Zumaia section (Batenburg et al., 2012) and provides a basic 405-kyr stratigraphy without introducing short eccentricity, obliquity or precession component features into the records. The age tie points for the 405-kyr age model are in the Supplement Table S1. To test the completeness of the Shatsky Rise record, we used the tie points of Batenburg et al. (2012)
updated for the Laskar cosine function for the Zumaia succession and plotted bulk δ¹³C records (Fig. S3, Table S2). The match of both records is remarkable, suggesting that both the Zumaia and the Shatsky Rise recorded global carbon cycle variations.

## 2.5 Scanning XRF Ba

Non-destructive X-ray fluorescence (XRF) data were collected every 2 cm down-core using XRF core scanner Avaatech serial no. 17 from the XRF core scanning facility at the IODP Gulf Coast Repository at Texas A&M University (College Station,
USA) in March 2017 for Sites 1209 and 1210 (Datasets S4-S6). Data were collected over a 1.2 cm² area with a down-core slit size of 10 mm using generator settings of 50 kV and a current of 0.75 mA, ideally for detecting Ba, and a sampling time of 12 s in each run directly at the split core surface of the archive half. The split core surface was covered with a 4 μm thin SPEXCerti Prep Ultralene foil to avoid contamination of the XRF detector prism and desiccation of the cores. The data were acquired with a SiriusSD 65mm² Silicon Drift Detector Model 878-0616B - SGX Sensortech with 133 eV X-ray resolution at 5.9keV
and 3kcps, the Canberra digital spectrum analyzer DAS 1000, and an Oxford Instruments 50W Neptune X-ray tube with a rhodium (Rh) target. Raw data spectra were processed using the Analysis of X-ray spectra by Iterative Least square software (bAxil) package from Canberra Eurisys. The scanning XRF Ba was used as a qualitative record of changes in biogenic barium flux at this open ocean setting after assessing minimal difference between XRF Ba and XRF Ba normalized to total measurements (XRF Ba/total area) as shown in Fig. S4. Additionally, XRF Sr/total area ratio (Fig. S5) is shown.

## 2.6 Composite Record

The scanning XRF Ba elemental intensities were used to correlate Maastrichtian cores from ODP Leg 198 Holes 1209A, 1209B and 1209C as well as Holes 1210A and 1210B to establish new composite records for Site 1209 and 1210 (Figs. S6-S8). Additionally, all core intervals outside the composite have been mapped to the composite to be able to place data from





those intervals on the new splice. All relevant tables for composite constructions are given in Datasets S7-S9 for Site 1209 and
Datasets S10-S12 for Site 1210. To form a complete Maastrichtian succession both sites had to be combined to cover gaps and
avoid drilling disturbed intervals. Correlation tie points using XRF Ba elemental intensities (Fig. S8) are given in Dataset S13.

### 2.7 Barite accumulation rate (BAR)

Barite separation was done on all 51 discrete sediment samples across 5 intervals (Dataset S14). The barite separation method,
which dissolves carbonates, organic matter, Fe-Mn oxyhydroxides, silicates, and fluorides to obtain a purified barite residue,
was modified from Paytan et al. (1996) as outlined in the Supplement Table S3. Specifically, acetic acid was used to slowly
(but completely) dissolve carbonates because the samples had such high carbonate content (> 96 wt %). Sample ashing in a
furnace to remove refractory organic material was done after the scanning electron microscope (SEM) observation due to the
low weight of the residue (and low organic material content). Previous work (Eagle et al., 2003) showed that the barite
separation method of Paytan et al. (1996) resulted in less than 5% loss of original barite when followed accordingly. Each
sample residue was then imaged using a Hitachi Benchtop SEM TM3030 at Kent State University to determine the average
percentage of barite (i.e., purity) by taking images of five random spots (Fig. S9). The purity was multiplied by the dry weight
of the residue and divided by the total dry bulk sample weight to determine the pelagic marine barite wt %.

Barite accumulation rate (BAR) was then calculated by multiplying the pelagic marine barite wt % by the mass
accumulation rate (MAR). This corrects any bias using wt % data that could result from varying (LSR) or changes in dry bulk
density (DBD). To determine the MAR, linear sedimentation rate (LSR) was multiplied by dry bulk density (DBD) from
Bralower et al. (2002). Dry density in each core section was used where available, such as for Core section 198-1210B-25H-
4. When there was no dry density measured for the specific core section, nearby dry density average was taken (1209B-27H-
6 and -31H-1 DBD were used for 1209B-29H-1 and -2 and 1209C-21H-1) or nearest DBD was adopted (1209A-26H-4 was
used for 1209C-16H; 1209B-31H-6 DBD was used for 1209C-22H- 2 and -3).

### 2.8 Excess-Ba (bio-Ba) accumulation rate (excess-BaAR)

Bulk digestion of aliquots of the same discrete bulk samples used for barite separation above was done to determine excess-
Ba (i.e., biogenic Ba) content from measurements of Ba and Al (Dataset S15). A modified bulk digestion method from Scudder
et al. (2014) was used as outlined in the Supplement Table S4 and the resulting solution was measured using a Perkin Elmer
3000DV ICP-OES in the Trace Element Research Lab at The Ohio State University. Table S5 shows the wavelength for each
150    element analyzed and relative standard deviation (2RSD), determined from replicate analyses of the medium standard solution
(n = 7). Measurements were 4% (2RSD) for Ba. Other elements (Ca, Fe, Al, Ti, Mn, Mg, Sr, and K) had 2RSD ranging from
2 to 3%, but Si had 2RSD of 16%.

Excess-Ba was calculated using an equation (Supplement Text S1) from Dymond et al. (1992) which corrects for any
contribution of non-biogenic Ba. For this study, terrigenous Ba/Al ratio (Ba/Al$_{terrigenous}$) of 0.0037 was used (Reitz et al., 2004).
Additional corrections for Ba associated with authigenic components and Fe-smectite using measurements of Mn and Fe



(Olivarez Lyle and Lyle, 2006) were not needed as these corrections resulted in differences of 6% or less (on average 2% difference, see Supplement Text S1). Excess-Ba accumulation rate (excess-BaAR) was then calculated by multiplying the excess-Ba (in ppm) by the MAR as determined above.

## 2.9 Benthic foraminiferal data

For quantitative studies of benthic foraminifera (Dataset S16) 16 selected samples were taken from the cores of Site 1209 spanning several cycles observed in scanning XRF Ba data. Samples of 10 to 15 cm$^3$ were taken at the Gulf Core Repository (GRC), then oven dried <50°C overnight and washed using tab water over a 63µm sieve to extract the sand sized fraction at MARUM Bremen. Wet sieved samples were oven dried <50°C overnight and weight to determine coarse fraction weight %. Benthic foraminiferal quantitative analysis was carried out at University of Zaragoza based on representative splits of

approximately 300 specimens larger than 63 µm. The percentages of individual taxa, species with agglutinated tests, and the Superfamily Buliminacea were calculated (Dataset S16). The Benthic Foraminiferal Accumulation Rates (BFARs) were calculated as a qualitative proxy for delivery of organic matter to the seafloor (Jorissen et al., 2007), using the number of benthic foraminifera per gram of sediment > 63 µm, the weight % of the sample > 63 µm, the linear sedimentation rate and the sediment density. BFARs are expressed as the number of benthic foraminifera (nr) accumulated per cm$^2$ and per kyr.

## 3 Results and discussion

### 3.1 Orbitally tuned continuous tropical Pacific record

Carbonate-rich, deep marine sediments drilled on the Shatsky Rise during Ocean Drilling Program (ODP) Leg 198 were used to create a new continuous composite record combining Sites 1209 and 1210 (Fig. 3). These sites were situated between 1500 and 2000 meters water depth in a productive upwelling region approximately 10ºN in the tropical Pacific during the

Maastrichtian (Fig. 1; Bralower et al., 2002). Previous studies on the Maastrichtian at Sites 1209 and 1210 identified the inoceramid acme and extinction, changes in ocean circulation, carbonate dissolution and episodes of higher productivity prior to the end-Cretaceous extinction event (Bralower et al., 2002; Frank et al., 2005; Alegret and Thomas, 2009; Jung et al., 2012, 2013; Dameron et al., 2017). However, prior age models and data resolution were incomplete and insufficient to identify precessional cycles in these open ocean tropical Pacific sedimentary records.

180       A highly resolved bulk carbonate δ$^{13}$C and δ$^{18}$O record was generated from a new composite record which shows a clear imprint of the stable 405-kyr eccentricity cycle (approximately 5-m cycle, Fig. S2, S10). The bulk carbonate δ$^{13}$C record thus was tuned to a 405 kyr long-eccentricity cycle target function from Laskar et al. (2004) and Laskar (2020) to form a basic age model from 71.5 to 66 Ma (Table S1). Only the 405 kyr eccentricity cycle function was employed because orbital solutions for the short eccentricity (100 kyr) cycle, which appears to be present in the record as an approximately 1-m cycle (Fig. S10),

cannot be used for intervals older than 60 Ma (Laskar et al., 2011). The tuned bulk δ$^{13}$C record is consistent with the existing





subtropical Atlantic high-resolution cyclostratigraphic record from Zumaia in northern Spain (Fig. S3; Batenburg et al., 2012, 2014) suggesting both records to be complete.

The bulk carbonate $\delta^{13}C$ values from Shatsky Rise closely resemble the lower resolution surface planktic foraminiferal $\delta^{13}C$ records from the same sites reported by Jung et al. (2013) and Dameron et al. (2017). Hence, the bulk carbonate isotope

record in the region primarily reflects surface conditions, which can be influenced by changes in local productivity (Fig. S5).

## 3.2 Export production proxies

Pelagic marine barite (Fig. S9) weight percent (wt %) and excess-Ba determined on the same samples between 71.5 to 66 Ma are positively correlated (Fig. 2), confirming that excess-Ba represents the biogenic or pelagic marine barite formed in the water column within microenvironments of decaying organic matter (Paytan et al., 1996). For the same or neighboring samples

(within 1 cm), X-ray fluorescence (XRF) Ba also has a strong positive correlation with both pelagic marine barite and excess-Ba. After calculating and comparing accumulation rates of barite (BAR) and excess-Ba (excess-BaAR) to scanning XRF Ba, the strong positive correlation remains (Fig. S11). This confirms that all barium records represent changes in export production in these sediments.

The equivalent barite accumulation from excess barite measurements, however, is higher than that determined from

extracting barite. Either non-barite (i.e., non-biogenic) Ba phases are included in the excess-Ba data biassing the data (Murray et al., 2000; Eagle et al., 2003; Gonneea and Paytan, 2006) or BAR is underestimating barite accumulation due to the mechanical loss of barite or its dissolution during sample processing, which is possible for samples with low barite wt % (Eagle et al., 2003). Such offsets between proxies are a concern when trying to precisely quantify variations in proxy measurements in terms of export production. Nonetheless, offsets are less of a concern when looking at one site over time and comparing

trends qualitatively to determine relative changes in export production, such as for this study.

Scanning XRF Ba was also compared with XRF Ba normalized to 'total area' (the sum of all counts for all elements processed from the 50 kV run data) (XRF Ba/total area) in Fig. S4 to confirm that there is little to no volume effect on scanning XRF Ba trends. Altogether, we conclude that XRF Ba is an ideal proxy to reconstruct export production at high resolution in the open ocean tropical Pacific at Sites 1209 and 1210 during the Maastrichtian.

## 210 3.3 Orbital cyclicity in the tropical Pacific biological pump

We observe orbital cyclicity in the continuous composite records of scanning XRF Ba and bulk carbonate $\delta^{13}C$ and $\delta^{18}O$ from Sites 1209 and 1210 (Fig. 3, and Figs. S10, S12-S13). While the 405 kyr long-eccentricity used to constrain the age model dominates bulk carbonate $\delta^{13}C$, it is weaker in bulk carbonate $\delta^{18}O$ and XRF Ba. This suggests that bulk carbonate $\delta^{13}C$ is primarily responding to factors other than temperature or salinity and organic C export which impact the bulk carbonate $\delta^{18}O$

and XRF Ba, respectively. The 100 kyr short-eccentricity cycle is weak in all three records and obliquity is not present in any of the records. Precession is observed in all records but is strongest in XRF Ba, demonstrating that precession, not the





modulating eccentricity cycle, played the major role in pacing organic C export or export production in the tropical Pacific during the Maastrichtian.

The variations in scanning XRF Ba throughout the composite record are interpreted as variations in export production and provide robust evidence for a tropical Pacific Ocean dominated by low-latitude (precessional) climate variations which responded strongly to precession even when the amplitude was low (e.g., during 100kyr eccentricity minima). When combined with existing records from the Atlantic during the Late Cretaceous (e.g., Herbert et al., 1990, 1999; MacLeod et al., 2001; Batenburg et al., 2012, 2014), a significant response of the Earth system due to the orbital forcing of precession is clear (Herbert et al., 1999; Clement et al., 2004).

The BAR and excess-BaAR records show export production increased by a factor of two to four within a single precessional cycle, in step with changes in scanning XRF Ba (Fig. 4, and Fig. S14). These large changes in the flux of organic matter falling through the water column can be caused by changes in surface production, transport efficiency, or both (Murray et al., 1996; Griffith et al., 2021). This data also suggests that these sites had generally low rates of new production (<2.2 gC cm$^{-2}$ yr$^{-1}$ using Dymond et al. (1992), see Supplement Text S2) or that a very small fraction of the surface production was

exported to depth (i.e., low transfer efficiency).

During these same precessional cycles, the barite related proxies are in phase with the amount of exported organic matter that reaches the seafloor (i.e., sequestration flux), reconstructed using benthic foraminifera accumulation rates (BFARs; e.g., Alegret et al., 2012, 2020; Griffith et al., 2021) (Fig. 4). A similar trend is shown in the percentage of the Superfamily Buliminacea, which in the absence of low-oxygen conditions, is indicative of enhanced food supply to the seafloor (Jorissen

et al., 2007; Alegret et al., 2012), and the percentage of the oligotrophic species *Oridorsalis umbonatus* is higher when Ba drops in the intervals of low export productivity (Fig. S15). These results suggest coupling between export production and carbon sequestration (i.e., carbon reaching the seafloor) in the tropical Pacific open ocean during the Maastrichtian. The variability in these short but high-resolution records is similar to that of lower resolution records (e.g., Frank et al., 2005), suggesting that caution is needed when interpreting lower resolution records of a highly variable signal, where drivers of

variability can be masked in the lower resolution records.

**3.4 Direct response to precession during greenhouse**

Carbon export from the surface ocean in the tropical Pacific (evidenced by XRF-Ba content) appears to have changed as a direct response to high frequency (~21 kyr) seasonal, insolation forced tropical precipitation. This fingerprint of precession in the deep-sea sedimentary record is most simply explained as the result of variations in surface productivity forced by cyclic

variation in water column stratification, upwelling intensity, and continental nutrient fluxes. In the tropical Pacific, where the thermocline depth has a strong influence on ocean primary production (Turk et al., 2001), a ~21-ky precession cycle could have been dominant in the Maastrichtian (Fig. 5). Shoaling of the thermocline would result in increased primary production and carbon export due to enhanced upwelling of nutrients, and this new record suggests such variations could have occurred in response to changes in precession. Previous work suggested a link between past warm climates and a "permanent El Niño"





state" (Fedorov et al., 2006) with a deeper thermocline and more stratified tropical Pacific with reduced primary production. The large range of variations in our record during the Maastrichtian greenhouse argues against this hypothesis. Instead, this new record adds to the growing body of work that suggests robust ENSO-like variability existed in past greenhouse conditions (e.g., Davies et al., 2012).

Reduced stratification and enhanced upwelling increase primary production and organic carbon export, but are expected
to induce a greater release of $CO_2$ to the atmosphere and act as a positive feedback to warming, unless the enhanced biological pump effectively suppresses the increase in $CO_2$ outgassing (e.g., Kim et al., 2019). Increased organic carbon export would also increase the corrosivity of deeper waters (e.g., Lyle et al., 1995), although $CaCO_3$ content in sediments remained high throughout this time at this site (>95%; Dataset S14). The maxima in corrosion-resistant benthic foraminifer (*N. truempyi*) and percent of agglutinated taxa, however, coincide with Ba maxima providing further support of increased organic carbon export
(Fig. S15).

The high frequency, precessional pacing of the carbon export in the tropical Pacific persisted throughout the Maastrichtian. This suggests it was a consistent, intrinsic feature of the end Cretaceous cool greenhouse, despite longer (and larger) scale changes, e.g., early Maastrichtian cooling pulse (EMCP; Haynes et al., 2020), warm mid-Maastrichtian event (MME; MacLeod and Huber, 1996), the early phase of Deccan Traps Volcanism (67.5 to 67.1 Ma; Chenet et al., 2007; Keller et al., 2016), or
the Latest Maastrichtian Warming Event (LMWE; Li & Keller, 1998; Gilabert et al., 2021). Organic carbon export appears largely decoupled from the abundance and eventual extinction of inoceramid bivalves related to the MME, suggesting that additional drivers (e.g., emplacement of large igneous provinces) played an important role in biotic change during this time. An increase in amplitude of the variations in carbon export recorded towards the end of the Cretaceous suggests an amplified sensitivity to orbital forcing when $CO_2$ was higher during Deccan volcanism, beginning around 67.5 Ma (Chenet et al., 2007).
As anthropogenic $CO_2$ emissions increase today, we need to consider whether or not sensitivity to seasonal insolation in the tropics might increase and dominate changes in the ocean-carbon-climate system in this key region.

**Data availability**. All data are available in the Supplement.

**Supplement.** The supplement related to this article will be available online at:

**Author contributions.** All authors contributed to the ideas expressed in this manuscript. TW initiated this project, which was further developed by EMG, and J-EK; TW and UR analyzed XRF and bulk carbonate isotopes; J-EK prepared and analyzed barite samples under the guidance of EMG; LA prepared and analyzed the benthic foraminiferal results. TW created the age



model and composite record. J-EK, EMG, TW and AJ did the artwork. All authors contributed to data interpretation and manuscript writing.

**Competing interests.** The contact author has declared that neither they nor their co-authors have any competing interests.

**Disclaimer.** Publisher's note: Copernicus Publications remains neutral with regard to jurisdictional claims in published maps and institutional affiliations.

**Acknowledgements.** Samples were provided by the International Ocean Drilling Program. We thank the staff at the IODP Gulf Coast Core Repository (GCR) for assistance at the Texas A&M University XRF Core Scanner Lab and sampling
guidance. We also thank Faizura Ahmad Zulkifli and Anis Hishammudin for assistance in barite separation, Samantha Carter for supervision on the separation process, Julia Sheets and Susan Welch for supervision on SEM sample coating, Anthony Lutton for supervision on ICP measurements, Merida Keatts at Kent State University for supervision on the SEM operation, and Jason Curtis at University of Florida for processing the carbonate weight percent.

**Financial support.** Funding for this research was provided by the Deutsche Forschungsgemeinschaft (DFG, German Research Foundation) to TW and UR - project no. 320221997, and by the Spanish Ministry of Economy and Competitiveness and FEDER funds (PID2019-105537RB-I00) to LA. Funding was also provided by the Deutsche Forschungsgemeinschaft (DFG, German Research Foundation) under Germany´s Excellence Strategy – EXC-2077 – 390741603.  This work was also supported in part by Ohio State Friends of Orton Hall to J-EK and US National Science Foundation grant OCE-1536630 to
EMG.

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





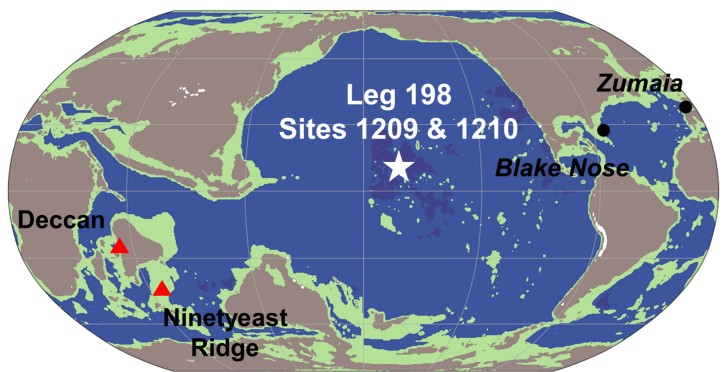


**Figure 1: Location of ODP Leg 198 Sites 1209 and 1210 at Shatsky Rise at 69 Ma (star) in the tropical Pacific (Scotese and Wright, 2018). Deccan and Ninetyeast Ridge are shown as red triangles. Location of Zumaia (Batenburg et al., 2012, 2014) and Blake Nose (MacLeod et al., 2001) records are shown as black circles.**

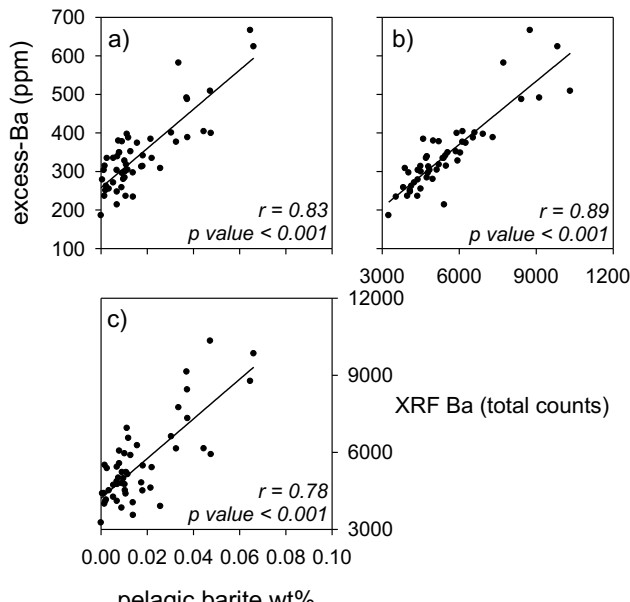


**Figure 2: Pelagic marine barite weight percent (wt %), excess-Ba (ppm), and XRF Ba (total counts) compared to each other. All correlations between barite related proxies are significant ($p < 0.001$).**

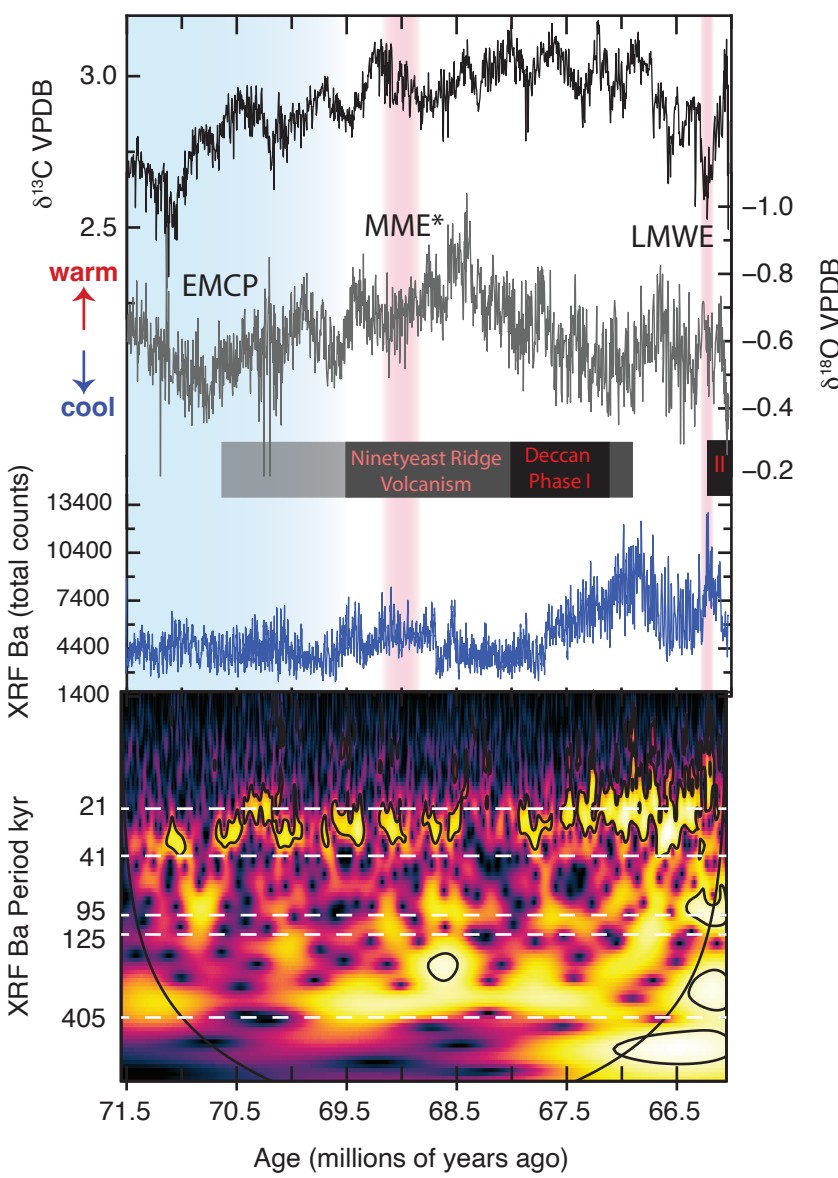

**Figure 3:** Bulk carbonate δ¹³C, δ¹⁸O, scanning XRF Ba, and wavelet analysis (Torrence and Compo, 1998; Grinsted et al., 2004) on non-stationary periodicities for XRF Ba (500 loess composite of Sites 1209 and 1210) show signals of precession in scanning XRF Ba which strengthen after Deccan volcanism begins. Brighter colours correspond to higher power within a corresponding frequency period (in thousands of years or kyr). Ninetyeast hotspot volcanism and approximate duration of Deccan volcanism shown, adapted from Keller et al. (2016). EMCP = early Maastrichtian cooling pulse (Haynes et al., 2020); MME* = mid-Maastrichtian event beginning at Chron C31n and lasting ~500kyr (Voigt et al., 2012); LMWE = latest Maastrichtian warming event (Li and Keller, 1998).



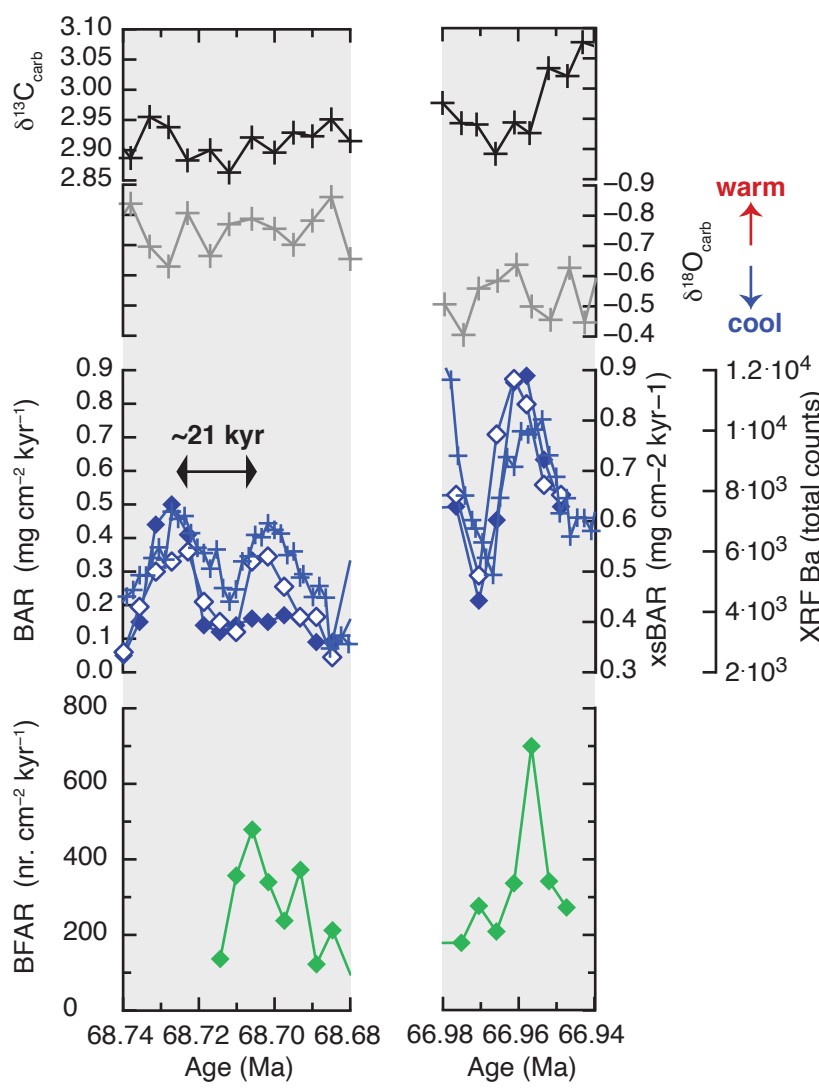


**Figure 4: Two discrete sample intervals covering potential cycles of precession (~21 kyr) with surface water related proxies (bulk δ¹³C, δ¹⁸O), organic carbon export proxies (scanning XRF Ba = crosses, excess-BaAR = open diamonds, and BAR = solid diamonds), and proxy for organic carbon arrival at the seafloor (BFAR = green diamonds). A strong positive correspondence is seen between carbon export from the surface of the oceans (XRF Ba, excess-BaAR, BAR) and organic carbon arrival at the seafloor (BFAR).**






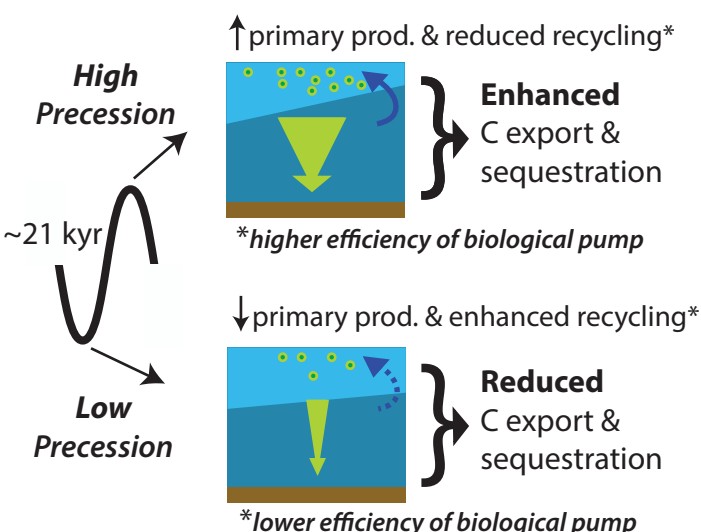

**Figure 5: Cycles of enhanced and reduced carbon (C) export and sequestration result from precessional paced variations in primary productivity and recycling in surface waters (i.e., changes in the efficiency of the biological pump). A shallower thermocline at high precession (steeper boundary between light and dark blue waters) would result in increased primary production and reduced recycling in the surface waters due to changes in water column stratification, continental nutrient fluxes and upwelling intensity which increases nutrients (solid blue arrow) and enhances C export out of the surface waters (wide green arrow). A deeper thermocline at low precession (near horizontal boundary between light and dark blue waters) would result in reduced primary production and enhanced recycling in the surface waters due to reduced nutrient availability (dashed blue arrow) and reduces C export out of the surface waters (thin green arrow). This ENSO-like variability could have controlled carbon export in the Maastrichtian.**