# Peer review of "Precessional pacing of tropical ocean carbon export during the Late Cretaceous"

_Climate of the Past, 2022_

## Referee Comment (RC2)

Kim et al. present a comprehensive study on the orbital influence on the tropical Pacific during the Cretaceous using the firsthand, high-resolution paleoclimate proxies. Time series analysis and the 405-kyr tuning of the d13C enable a high-resolution astrochronology allowing the evaluation of the dominant cyclicities of the export production at an unprecedented level. The choice of the proxies for the organic carbon export is reasonable. This study provides an excellent example of how to explore the driving force in the tropical biological pump. Therefore, this study is of high quality and publishable after addressing the following issues.

1. Missed information about the tuning in the main paper

Astrochronology is the basis of the whole story. Unfortunately, the cyclostratigraphic results are only presented in the Appendix. One has to go to the supplementary figures to get information such as power spectra, determination of the sedimentation rate, filter, and tuning strategies. Therefore, the structure of this paper needs an improvement to enhance its readability. For example, Figure S2 should appear in the main paper. Also, it would be good to see (at a minimum of one pair of) the power spectrum and wavelet analysis in the main paper but this should be decided by the authors.

Moreover, the astronomical tuning strategy is generally straightforward in this case study. Assuming that the 5 m wavelengths represent long eccentricity cycles, a mean sedimentation rate of about 1.25 cm/kyr is supported by other geologic evidence. However, the power ratio method can be subjective and it is unclear whether all assigned 405 kyr cycles are fine or not. One solution is to add results using more objective statistical tuning approaches, such as ASM, TimeOpt, and COCO. As you can see below, quick COCO/eCOCO analyses of your data strongly support the robustness of results presented in this manuscript, although the choice of 405 kyr cycles (for example, at ~285 m) needs more discussion.

[Figure]

Figure 1. The composite d13C data series has been interpolated using a median sampling rate of 0.05 m. A 10 m lowess trend was removed. The COCO method suggests the most likely sedimentation rate is at ~1.3 cm/kyr. Multiple peaks suggest the sedimentation rate is variable as shown in the following eCOCO sedimentation rate map.

eCOCO analysis using an 8 m sliding window demonstrates the evolution of the sedimentation rate, which generally supports your results, but does show a minor discrepancy at ~285 m.

[Figure]

**2. Unclear relationship between the thermocline and precession**

This paper argues that precession paced ocean export production via influencing the thermocline depth. In the text, shoaling of the thermocline is thought to have occurred in response to changes in precession. This is a critical component of the story; however, the mechanism is not presented. In Figure 5, the authors note that "a shallower thermocline at high precession" represents a huge jump. Please fulfill this gap in the main text.

Moreover, the paper mentioned that "ENSO-like variability" existed in past greenhouse conditions. It would be better if this variability pattern can be introduced so readers don't have to read other papers (e.g., Davies et al., 2012) to understand this term.

**Minor issues**

1. Line 28: add "405 kyr," before 100 kyr.

2. Line 111: sampling time of 12 s. Do you have evidence that 12 s is sufficient for the data quality? I have tested a range of sampling times ranging from 10 s to 120 s, it looks like XRF can usually produce stable results after ca. 20 s.

3. Figure 1: Please explain the meaning of shade zones in the map (brown, light olive, blue, and purple).

4. Figure 3: obliquity period was not 41 kyr. Alternatively, the La2004 solution predicts a 38 kyr cyclicity for this time interval.

5. Figure 4: the middle rows are too busy. Why there is no BFAR data at the interval of 66.74-68.71 Ma?

6. Line 468: Remove "Strong" in the caption.

Mingsong Li

---

## Author Response (AR1)

**THE OHIO STATE UNIVERSITY**

Dr. Elizabeth M. Griffith,
Assoc. Prof., *corresponding author* griffith.906@osu.edu
Mobile: +1(330)-285-0441
School of Earth Sciences, The Ohio State University
125 South Oval Mall, Columbus, OH 43210 USA

22 September 2022

We thank the editor for the opportunity to revise the manuscript for consideration of publication in *Climate of the Past.* We have included a response to the reviewers' comments here and highlighted in the tracked change documents that follow where the manuscript and supplement was revised.

Note the datasets are provided only for invited reviewers through the password protected site (we do not want to have the full dataset available for the public until the manuscript is published): Editor/Reviewer only password for Dataset: GriFFWestH198

Best regards,

Dr. Elizabeth M. Griffith
* * *
11 Aug 2022

Editor decision: Reconsider after major revisions
by Zhengtang Guo

**Comments to the author**:
Dear Elizabeth M. Griffith,

Thank you for submitting your work to Climate of the Past. I have completed the review of your manuscript and the referees' reports are on the web. As you had seen, both referees indicated the significance of your study for understanding the orbital forcing on the tropical ocean carbon export during the Late Cretaceous. They raised, however, some important issues requiring your considerations through a significant revision. I am not intending to repeat their comments because they all are clear and constructive. I invite you to resubmit the manuscript after addressing all reviewer comments. The revised version will be sent to the referees again.

I am looking forward to receiving the revised manuscript.

All the bests
Zhengtang GUO

Referee comment on "Precessional pacing of tropical ocean carbon export during the Late Cretaceous" by Ji-Eun Kim et al., Clim. Past Discuss., https://doi.org/10.5194/cp-2022-42-RC1, 2022

The Shatsky Rise in the Northwest Pacific preserves high-quality deep-sea sediment spanning the whole period of the Cretaceous and the Cenozoic, being an ideal place to perform studies in the field of paleoceanography and paleoclimate. The previously wellknown research work has been focused on the Cenozoic. However, the research works in the Cretaceous are all based on low time resolution proxy records that limits our understanding on a warmer than today climate condition.

Kim and co-authors presented a high-resolution composite record from ODP Leg 198 Sites 1209 and 1210 on Shatsky Rise, Northwest Pacific during the Maastrichtian Stage from the Upper Cretaceous Epoch that lasted from 71.5 to 66 Ma. The new dataset includes the high time resolution (3-4 kyr) bulk carbonate d13C and d18O measured in 1394 samples and ultra-high time resolution (~1 kyr) non-destructive X-ray-fluorescence (XRF) Core Scanning Barium data collected every 2 cm. For the first time, this high-quality and high time resolution dataset allows us to perform paleoceanographic studies on orbital time scale, probing the imprints of orbital forcing in the deep sea sediment of the Pacific Ocean during a warmer than today or greenhouse climate state.

The paper is well written with clear structure, which is easy to be understood. The supplementary information includes enough and detailed materials for evaluation. In summary, the newly presented dataset in the Shatsky Rise has great potential for improving our understanding on the forcing mechanism of the orbital scale climate change in a greenhouse world. I would encourage publication in Climate of the Past if they could consider a few of my major concerns as listed below.

The key point of this manuscript is "precessional pacing". However, spectral analyses in both depth (independent of the tuned age model based on comparison of 405 ka cycle) and time (Figures S10, S12 and S13) display that the most prominent orbital cycle in the variability of planktonic d13C and d18O and the Ba is the 405 ka long eccentricity cycle, with strong 21 ka precession cycle, weak 41 ka obliquity cycle and relatively weak 100 ka short eccentricity cycle. The planktonic foraminiferal d13C is closely related to the biological pump and the carbon export production of the upper ocean. The Eccentricity modulates climate change through a nonlinear forcing because the eccentricity's contribution to insolation is too small to take effect in a linear insolation-climate system. The 405 ka long eccentricity cycle is also the most remarkable and stable orbital cycle for the past 250 million years. The 100 kyr short eccentricity cycle also modulates the amplitude of the climate precession. Therefore, why "the precessional pacing of tropical ocean carbon export"?

> Based on the spectral analysis given in the original Figures S10, S12 and S13 the picture is more complex for each proxy data. Clearly, as given in the manuscript and used to establish the 405-kyr stable cyclostratigraphy, the bulk carbon stable isotope data are dominated by the long eccentricity cycle (highest peak in the MTM – revised Fig. 2). A feature well known from the Cenozoic, very likely related to the long residence time of carbon in the ocean (See Pälike et al. 2006, Science). For the Late Cretaceous our record is the first that clearly shows the 405-kyr dominance in bulk carbon isotope data. The bulk oxygen isotope data (now Fig. S11) do not show this strong 405-kyr influence but more power in the MTM spectra in the higher frequencies related to precession. One important point looking at age MTM spectra is to be aware that the basic age model is a 405-kyr cyclostratigraphy which will enhance the long eccentricity related spectral peaks. Looking at the MTM spectra in the depth domain is much less biased. However, there is a strong 405-kyr component in both carbon and oxygen bulk isotope data. We are cautious to interpret the oxygen bulk stable isotope data because it will be a mix of calcareous nannofossils from the surface ocean, planktonic foraminifera from the upper ocean layers and, to a lesser extent, the deep sea benthic foraminifera. It is not focus of this manuscript.
>
> But the barium elemental data, which is focus of the manuscript, shown in the MTM spectrum in now Fig. S12 clearly is dominated by short cycles in the order of 20-30 cm and a really minor component of a 5m cycle, which is equivalent to the long eccentricity cycles if the short cycles are precession (~20 precession cycles in one 405-kyr cycle; ~20 25cm cycles in a 5 m cycle). Even with the 405-kyr age model the MTM spectrum clearly shows the domination of precession. Looking at now Fig. 5 of the main text (this Figure needs to be full page as shown in the revised version to better illustrated the fine cyclicity) and the now main Fig. 3 as well as the correlation now Fig. S6 there is one dominating rhythm related to precession with very little expression of modulations by eccentricity. Prominent examples of eccentricity modulated precession cycles in geochemical data is the late Paleocene and early Eocene (see Lourens et al. 2005; Westerhold et al. 2007 and 2008) are published from Walvis Ridge in the South Atlantic. Compared to those records the Maastrichtian Shatsky Rise XRF records show a minor eccentricity component. Because of the modulation of the precession by eccentricity some related variation in amplitude of the

data can be expected and is seen, but not to an extent that would lead to the interpretation of major importance on the data.

In the manuscript we focus on the cyclicity of XRF Barium data which we show are very likely related to changes in surface ocean productivity. And because these data are dominated by precession our manuscript focuses on the precessional pacing of tropical ocean carbon export. We think this is justified by the outstanding quality of the data.

No changes were made to the manuscript following this comment.

The traditional concept of "biological pump" was cited to interpret the ocean carbon export production in this study. In Figure S5, the authors compared the high time resolution bulk d13C and d18O with the low time resolution planktonic foraminiferal d13C and d18O. They concluded that the two kinds of d13C records resemble to each other. As far as I am concerned, only the bulk carbonate d13C shows similarity to the Rugoglobigerina rugosa (subsurface species, deeper) d13C (Figure S5, top, green) but big differences with the Pseudoguembelina costulata and P. kempensis (surface species, shallower) d13C (Figure S5, top, orange). In the greenhouse world of the Cretaceous, a time interval without global ice volume effect on the calcareous shells of planktonic foraminifers, the temperature and salinity are two major factors controlling the planktonic foraminiferal d18O. As seen in Figure S5, the d18O of the subsurface species (green) are obviously heavier than that of the surface species (orange), indicating cooler or saltier water mass. However, the d13C of the subsurface species (green) is also heavier than that of the surface species (orange), which is opposite to the vertical distribution of the water mass d13C caused by the traditional biological pump that decreases with water depth. The inconsistency probably indicates that the ocean carbon export production of the tropical Pacific Ocean during the greenhouse world of the late Cretaceous is different with that we have known today. The authors need to add a new paragraph to discuss this inconsistency before they could use the XRF core scanning Ba record as a proxy of the ocean carbon export production.

Explaining this apparent inconsistency that the reviewer outlines in the foraminiferal isotope data does not preclude using the XRF scanning Ba record as a proxy of ocean carbon export production since the Ba proxy is independent of the foraminiferal and bulk C isotope data being that it relies on the modern observations that the mineral barite accumulates under regions of high organic carbon export can be used to independently reconstruct changes in organic carbon export following previous work in the Cenozoic (see Dymond et al., 1992; Paytan et al., 1996; Eagle et al., 2003; Griffith et al., 2021). The XRF scanning Ba record is confirmed in our study as a proxy for the mineral barite by carefully separating barite from discrete sediment samples (see Figs. 4, S7, S10). This is necessary to be sure that the XRF scanning record is actually recording marine pelagic barite and not a diagenetic phase or some other Ba-phase.

The relevance of the foraminiferal isotope records is in better understanding the dynamics in the water column – as the reviewer points out – and the primary influence on the bulk carbonate d13C record presented in high resolution for the first time in this study. Whether or not Rugoglobigerina rugosa represents surface or subsurface waters is less clear. Frank et al. (2005) suggested that R. rugosa occupied the mixed layer depth and contained photosymbionts (Abromovich et al., 2003) and was the most enriched in 13C followed by two other planktic foraminifera measured (G. stuartiformis and P. multicamerata) – similar to our presentation of R. rugosa with P. costulata and P. kempensis all of which have been suggested as mixed layer dwellers (see Abramovich et al., 2003). This suggests that P. costulata and P. kempensis could have been somewhat deeper dwelling than R. rugosa at this time with a surface d13C gradient consistent with expectations of the heaviest d13C at the shallowest depths. Jung et al. (2013) suggested that their planktic R. rugosa d18O record (plotted in Fig. S3) might have been altered toward cooler temperatures during early diagenetic recrystallization taking place on the seafloor – or as Reviewer 1 suggested R. rugosa could be recording a saltier water mass.

We thought it was important to see if the d13C variations we measured in the bulk carbonate record at high resolution could be related to surface production – and found

that they were generally similar to the planktic R. rugosa d13C record suggesting that they could be recording changes in local surface productivity. So we revised the original statement moved it to the proxy results and discussion section "Orbital cyclicity in the tropical Pacific biological pump".

"Because the bulk carbonate $\delta^{13}C$ values closely resemble the lower resolution surface planktic foraminiferal $\delta^{13}C$ record of *Rugoglobigerina rugosa* (Fig. S3) from the same sites reported by Jung et al. (2013) , it is possible that it reflects surface conditions potentially related to local surface productivity. However additional high resolution work is needed to confirm this initial observation."

The figure caption for the supplement also outlines what is mentioned above: "Figure S3. (from top to bottom) Shatsky Rise composite bulk δ13C (black line) with 50 points moving average (red line) shown with planktic foraminifera δ13C from Rugoglobigerina rugosa (Jung et al., 2013; green circles) and planktic foraminifera δ13C from Pseudoguembelina costulata and P. kempensis (Clark et al., 2012; Dameron et al., 2017; orange circles). All three were thought to have occupied the mixed layer depth (Abromovich et al., 2003), however given that P. costulata and P. kempensis have lower δ13C values, they likely were somewhat deeper dwelling than R. rugosa at this time if the surface δ13C gradient was consistent with expectations of the heaviest d13C at the shallowest depths. Shatsky Rise composite bulk δ18O (black line) with planktic foraminifera δ18O. XRF Ba (total counts) with 100 points moving average (red line). At the bottom, XRF Sr/total area is plotted with planktic:benthic foraminifera ratio (P:B ratio; Dameron et al., 2017) in green. Note age is decreasing from the left to right."

New reference added to Supplement: Abramovich, S., G. Keller, D. Stuben, and Z. Berner (2003), Characterization of late Campanian and Maastrichtian planktonic foraminiferal depth habitats and vital activities based on stable isotopes, Palaeogeogr. Palaeoclimatol. Palaeoecol., 202, 1–29.

The last part of this manuscript "3.4" focuses on the discussion of direct response to precession during greenhouse world. An important content of this part is the discussion on whether the tropical Pacific was more like a permanent El Niño like state or robust ENSO like variability existed in past greenhouse conditions. ENSO is El Niño-southern oscillation. The authors need to explain the difference in "El Niño like state" and "robust ENSO variability" in a greenhouse condition. Today, we usually use the changes of the gradients in both the SST (Sea Surface Temperature) and the Thermocline Depth of the east and west equatorial Pacific to depict the ENSO variability that is a typical climate phenomenon in the equatorial Pacific Ocean. The sites 1209 and 1210 were in the middle of the tropical open ocean in the late Cretaceous (Figure 1). Can we depict the ENSO variability if it really existed in the late Cretaceous without reconstructions of the gradients of the SST or thermocline depth in the east and west equatorial ocean? If not, the vague discussion based on non-proxy-derived discussion would lead to misunderstanding.

Yes, we agree with the reviewer that in order to reconstruct ENSO variability at this time, more than one record is needed. However, we argue that if there was a permanent El Nino like state at this time in the Pacific (with deeper thermocline and more stratified tropical Pacific) as previously hypothesized for past warm climates, we would not see the large variations in export production at our tropical Pacific site on relatively short timescales, i.e., precessional changes of more than 2x in export production and more than 4x in benthic foraminiferal accumulation rates.

This part of the manuscript was clarified and references added in response to this reviewer's comment and that by the second reviewer. Please see response that follows.

Referee comment on "Precessional pacing of tropical ocean carbon export during the Late Cretaceous" by Ji-Eun Kim et al., Clim. Past Discuss., https://doi.org/10.5194/cp-2022-42-RC2, 2022 – signed by Mingsong Li

Kim et al. present a comprehensive study on the orbital influence on the tropical Pacific during the Cretaceous using the firsthand, high-resolution paleoclimate proxies. Time series analysis and the 405-kyr tuning of the d13C enable a high-resolution astrochronology allowing the evaluation of the dominant cyclicities of the export production at an unprecedented level. The choice of the proxies for the organic carbon export is reasonable. This study provides an excellent example of how to explore the driving force in the tropical biological pump. Therefore, this study is of high quality and publishable after addressing the following issues.

1.   Missed information about the tuning in the main paper

Astrochronology is the basis of the whole story. Unfortunately, the cyclostratigraphic results are only presented in the Appendix. One has to go to the supplementary figures to get information such as power spectra, determination of the sedimentation rate, filter, and tuning strategies. Therefore, the structure of this paper needs an improvement to enhance its readability. For example, Figure S2 should appear in the main paper. Also, it would be good to see (at a minimum of one pair of) the power spectrum and wavelet analysis in the main paper but this should be decided by the authors.

Moreover, the astronomical tuning strategy is generally straightforward in this case study. Assuming that the 5 m wavelengths represent long eccentricity cycles, a mean sedimentation rate of about 1.25 cm/kyr is supported by other geologic evidence. However, the power ratio method can be subjective and it is unclear whether all assigned 405 kyr cycles are fine or not. One solution is to add results using more objective statistical tuning approaches, such as ASM, TimeOpt, and COCO. As you can see below, quick COCO/eCOCO analyses of your data strongly support the robustness of results presented in this manuscript, although the choice of 405 kyr cycles (for example, at ~285 m) needs more discussion.

[Figure]

Figure 1. The composite d13C data series has been interpolated using a median sampling rate of 0.05 m. A 10 m lowess trend was removed. The COCO method suggests the most likely sedimentation rate is at ~1.3 cm/kyr. Multiple peaks suggest the sedimentation rate is variable as shown in the following eCOCO sedimentation rate map.

eCOCO analysis using an 8 m sliding window demonstrates the evolution of the sedimentation rate, which generally supports your results, but does show a minor discrepancy at ~285 m.

[Figure]

R#2 suggests adding a chapter about the astrochronology / cyclostratigraphy in the main text as this is a central part of the study. In a revised manuscript we added this chapter dealing with the age model development. We move supplementary figures S2 (Basic 405 kyr age mode) and S10 (bulk carbon isotope MTM power spectrum and wavelet analysis in depth and age) to the main text.

R#2 points out that *the astronomical tuning strategy is generally straightforward in this case study* and that *the resulting mean sedimentation rate of about 1.25 cm/kyr* from the 405-kyr cyclostratigraphy is supported by other geologic evidence.

We have tested our 405-kyr cyclostratigraphy by plotting the Shatsky Rise bulk δ13C record against the bulk δ13C record of the Zumaia succession from Batenburg et al. (2012) which is the only record with comparable resolution. As given in the main text we updated the tie points of Batenburg et al. (2012) to the Laskar cosine function (Table S4) to make the records consistent with respect to the target curve. This way we can check if the number of 405-kyr cycles identified in the Shatsky Rise record is consistent with the complete record from Zumaia.

R#2 comments that the method we used *can be subjective and it is unclear whether all assigned 405 kyr cycles are fine or not*. One solution is to use complex statistical tools like TimeOpt or COCO, where the latter was developed by R#2. And indeed as given by the test by R#2 using COCO the results are basically the same as ours.

We thank R#2 for point out to use these methods. Moreover, R#2 suggests a potential critical interval identified by his method at ~285m rmcd. This corresponds to between 405-kyr cycles labeled Maa5 and Maa6 in what are now Figs. 3 and S8. If there is one more 405-kyr cycle in this interval the carbon isotope minimum at 68.3 Ma would not match the Zumaia record anymore.

In the revised version of the manuscript we added the COCO approach for testing our age model. It should be pointed out here that the drill cores used are affected by coring disturbance to some extent, particularly when chert layers were encountered. For the composite record from Shatsky Rise we tried to avoid those intervals whenever possible. However, there will be some disturbance and therefore misinterpretation of changes in cycle thickness can occur using an automated analysis routine that assumes constant

sedimentation rates. Unpublished precession cycle by precession cycle correlation to Zumaia also reconfirms the basic 405-kyr age model presented in the initial manuscript.

2.   Unclear relationship between the thermocline and precession

This paper argues that precession paced ocean export production via influencing the thermocline depth. In the text, shoaling of the thermocline is thought to have occurred in response to changes in precession. This is a critical component of the story; however, the mechanism is not presented. In Figure 5, the authors note that "a shallower thermocline at high precession" represents a huge jump. Please fulfill this gap in the main text. Moreover, the paper mentioned that "ENSO-like variability" existed in past greenhouse conditions. It would be better if this variability pattern can be introduced so readers don't have to read other papers (e.g., Davies et al., 2012) to understand this term.

> We added additional introduction to this variability in the revised manuscript (beyond what was done to address the comments from reviewer 1) and more clearly lay out the proposed relationship between precession and this ENSO-like variability (revisions in red) "…Previous work suggested a link between past warm climates and a "permanent El Niño state" (Wara et al., 2005; Fedorov et al., 2006) with weak trade winds along the equator, a deeper thermocline and more stratified tropical Pacific preventing upwelling of nutrients, which resulted in reduced primary production and low carbon export. Today this scenario occurs as an irregular periodic variation along with an opposite "La Niña" state in the tropical Pacific with strong trade winds and more intense upwelling in the eastern tropical Pacific driving increased productivity on interannual-to-decadal timescales. Changes in the mean state over longer timescales can be recorded in deep sea sediment archives (e.g., Pena et al., 2008; Zhang et al., 2021). The large range of variations in our carbon export record in the tropical Pacific (XRF-Ba elemental intensities and benthic foraminiferal accumulation rates) during the Maastrichtian greenhouse argues against the hypothesis of a continual El Niño-like state. The new record exhibits variations which could reflect changes in the mean climate state. We suggest that this new record adds to the growing body of evidence that suggests robust El Niño-Southern Oscillation-like variability existed in past greenhouse conditions (e.g., Davies et al., 2012) and this ENSO-like variability may be sensitive to orbital forcing, especially the effect of orbital precession in the tropics (e.g., Clement et al., 1999; 2000; Lu et al., 2019). The resulting tropical mean climate state and climate variability forced by minima and maxima of precession changing the strength of coupled ocean-atmosphere feedbacks in the tropical Pacific is less clear during the Maastrichtian greenhouse (Fig. 7), and requires focused modelling efforts and new proxy records to test these hypothesized relationships."

> Upon further examination and reflection of the literature, we want to shift the focus of the discussion here (see above). Figure 7 is now revised so that a precession is more generally related to changes in the mean state of the system, shifting from a mean state with a shallower thermocline resulting in higher productivity to a deeper thermocline with lower productivity in the region. Without additional work (outside the scope of this study) it is not possible to know if it is minima or maxima in precession that should result in one mean state or the other.

> New references were added to the main manuscript:
> Clement, A. C., Seager, R., and Cane, M. A.: Orbital controls on the El Niño/Southern Oscillation and the tropical climate, Paleoceanography, 14, 441– 456, 1999. doi:10.1029/1999PA900013.
> Clement, A. C., Seager, R., and Cane, M. A.: Suppression of El Niño during the mid-Holocene by changes in the Earth's orbit, Paleoceanography, 15, 731– 737, 2000. doi:10.1029/1999PA000466.
> Lu, Z., Liu, Z., Chen, G., and Guan, J.: Prominent precession band variance in ENSO intensity over the last 300,000 years, Geophysical Research Letters, 46, 9786-9795, 2019. doi:10.1029/2019GL083410.

Pena, L. D., Cacho, I., Ferretti, P., and Hall, M. A.: El Niño-Southern Oscillation-like variability during glacial terminations and interlatitudinal teleconnections, Paleoceanography, 23, PA3101, 2008. doi:10.1029/2008PA001620

Wara, M. W., Ravelo, A. C., and Delaney, M. L.: Permanent El Niño–like conditions during the Pliocene warm period, Science, 309, 758–761, 2005. doi:10.1126/science.1112596.

Zhang, S., Yu, Z., Gong, X., Wang, Y., Chang, F., Lohmman, G., Qi, Y., and Li, T.: Precession cycles of the El Niño/Southern oscillation-like system controlled by Pacific upper-ocean stratification, Communications Earth & Environment, 2, 239, 2021. doi:10.1038/s43247-021-00305-05.

Minor issues

Line 28: add "405 kyr," before 100 kyr.

Revised as suggested.

Line 111: sampling time of 12 s. Do you have evidence that 12 s is sufficient for the data quality? I have tested a range of sampling times ranging from 10 s to 120 s, it looks like XRF can usually produce stable results after ca. 20 s.

The Shatsky Rise cores consist of sediments with very high carbonate content, typically >95%. XRF scanning of very high carbonate sediments is difficult due to artifacts produced by the extremely high calcium peak and interaction with other elements. Thus, after test scans, T. Westerhold decided to scan only at a 50 kV setting because only the barium signal gave a stable and good result using 12 seconds count time. Agreed that count time should be longer to get statistically better results, however there is a time and cost limitation running cores at the repositories. Thus 12 seconds only scanning 50 kV was deemed useful to establish a stratigraphy and look at productivity related changes. Other elements like iron, which normally are pretty robust in acquiring, are difficult to measure on the high carbonate cores in relatively short time. One has to be aware that >95% carbonate content at a 10 kV run (usually the one to measure Al to Fe) will result is a high deadtime prolonging the scan time enormously. From the experience of T. Westerhold with many XRF scanned sections on a large range of sediment types and cores, stable results can be obtained already at 5-7 seconds count time having the right sediment composition and sediment core quality (not too high water content etc.). Generally, it cannot be stated as done by R#2 that XRF can usually produce stable results after ca. 20 s only.

Figure 1: Please explain the meaning of shade zones on the map (brown, light olive, blue, and purple).

The colors used for shading are now specified in the figure caption. We added "**Shading indicates land above 0 m (brown), land above 6 km (white), and ocean settings with water depths greater than 4km (dark blue), between 4km and 120 m (blue), and above 120 m (green).**"

Figure 3: obliquity period was not 41 kyr. Alternatively, the La2004 solution predicts a 38 kyr cyclicity for this time interval.

Because La2004 is not reliable at 70 Ma, we would prefer to just keep the 41 kyr indication as it is around where obliquity should be. No corrections were made following this comment.

Figure 4: the middle rows are too busy. Why there is no BFAR data at the interval of 66.74-68.71 Ma?

BFAR was only done for two cycles (peaks in Ba) to confirm whether or not the organic matter export indicated by the Ba peaks was reaching/impacting the seafloor community. No data was collected for their earlier peak.

Line 468: Remove "Strong" in the caption.

"Strong" was removed from caption.

---

## Referee Report (RR1)

The authors thoroughly considered nearly all the comments and suggestions of mine and the other reviewer's and made revisions accordingly. My major comment on the revised manuscript focuses on the discussion on the orbital cycle. As clearly presented in this study, the 405-kyr cycles in all the late Cretaceous proxy records from the tropical Pacific Ocean leave great impression on me. In the revision, however, the authors insisted on taking the precession as the dominant orbital forcing on the XRF-Ba changes (they use it as proxy of export productivity) during the late Cretaceous in the tropical Pacific Ocean, obviously ignoring the eccentricity's role (particularly the 405-kyr long eccentricity cycle) in modulating the hydrological cycle and productivity related carbon cycles. As clearly seen in all the spectral analyses in depth domain (no tuning effects; figure 2, figure S11, S12), both the bulk isotopes and XRF-Ba records display significant 5 m cycles that correspond the 405 kyr long eccentricity cycle according to their age model. All the wavelet analyses of the three proxy records show the same spectral features with the 405 kyr as the strongest and the most continuous cycle. Even though this paper focuses on the XRF-Ba derived productivity record, as they stressed in the rebuttal letter, the most abundant and significant spectral peaks that range from 0.025 to 0.06 (cycles/kyr) include the cycles of both the precession and nonprecession bands (figure S12; Please check the MTM spectral analyses in b. Is the unit cycles/meter of the X-axis in b as the same as in a?). Thus, what do these non-precession cycles represent? These non-precession cycles are as significant as the 19-23 kyr precession cycles. We have no doubt that the precession plays an important role in modulating the XRF-Ba derived productivity changes. However, the spectral and wavelet analyses also tell us that the other orbital cycles including those near the precession band and the 405-kyr long eccentricity cycle also play significant roles. This is the reason why I commented on the original draft that why do you only concentrate on the precession band? The CENOGRID climate records from the tropical oceans (Westerhold et al., 2020, Science), a great work led by one of the corresponding authors of this manuscript, display dominant 405-kyr cycles in the hydrological and carbon cycles throughout the whole Cenozoic, which is also one of the focuses of this famous paper. Was the 405 kyr cycle not as important as the precession in the late Cretaceous? At least, you should point out its role and clarify its relationship with the precession in the late Cretaceous tropical Pacific Ocean rather than ignore it and made no change in the revision.

---

## Author Response (AR2)

THE OHIO STATE UNIVERSITY

Dr. Elizabeth M. Griffith,
Assoc. Prof., *corresponding author* griffith.906@osu.edu
Mobile: +1(330)-285-0441
School of Earth Sciences, The Ohio State University
125 South Oval Mall, Columbus, OH 43210 USA

2 November 2022

We have made the minor revisions requested by the anonymous reviewer and have included a response to the reviewers' comments here and highlighted in the tracked change documents that follow where the manuscript and supplement was revised.

Note the datasets are provided only for invited reviewers through the password protected site (we do not want to have the full dataset available for the public until the manuscript is published): Editor/Reviewer only password for Dataset: GriFFWestH198

Best regards,

Dr. Elizabeth M. Griffith
* * *
23 October 2022

Editor decision: Publish subject to minor revisions (review by editor)
by Zhengtang Guo

Dear Drs. Ji-Eun Kim and Thomas Westerhold,

Thank you for submitting the revised version of your work. The reviewers have read it again. They are in overall satisfactory with the revisions you made. Reviewer#2 request adding some further explanations about the roles of the other orbital parameters in modulating the late Cretaceous climate. It would be finer if you can consider this suggestion in an appropriate way. Once you consider this minor point in a further revised version, I'll be happy to accept you work for publication in CP.

Thank you in advance.

With the best wishes

Sincerely,

Zhengtang GUO

Referee comment: The authors thoroughly considered nearly all the comments and suggestions of mine and the other reviewer's and made revisions accordingly. My major comment on the revised manuscript focuses on the discussion on the orbital cycle. As clearly presented in this study, the 405-kyr cycles in all the late Cretaceous proxy records from the tropical Pacific Ocean leave great impression on me. In the revision, however, the authors insisted on taking the precession as the dominant orbital forcing on the XRF-Ba changes (they use it as proxy of export productivity) during the late Cretaceous in the tropical Pacific Ocean, obviously ignoring the eccentricity's role (particularly the 405-kyr long eccentricity cycle) in modulating the hydrological cycle and productivity related carbon cycles. As clearly seen in all the spectral analyses in depth domain (no tuning effects; figure2,

figure S11, S12), both the bulk isotopes and XRF-Ba records display significant 5 m cycles that correspond the 405 kyr long eccentricity cycle according to their age model. All the wavelet analyses of the three proxy records show the same spectral features with the 405 kyr as the strongest and the most continuous cycle. Even though this paper focuses on the XRF-Ba derived productivity record, as they stressed in the rebuttal letter, the most abundant and significant spectral peaks that range from 0.025 to 0.06 (cycles/kyr) include the cycles of both the precession and non-precession bands (figure S12; Please check the MTM spectral analyses in b. Is the unit cycles/meter of the X-axis in b as the same as in a?). Thus, what do these non-precession cycles represent? These non-precession cycles are as significant as the 19-23 kyr precession cycles. We have no doubt that the precession plays an important role in modulating the XRF-Ba derived productivity changes. However, the spectral and wavelet analyses also tell us that the other orbital cycles including those near the precession band and the 405-kyr long eccentricity cycle also play significant roles. This is the reason why I commented on the original draft that why do you only concentrate on the precession band? The CENOGRID climate records from the tropical oceans (Westerhold et al., 2020, Science), a great work led by one of the corresponding authors of this manuscript, display dominant 405-kyr cycles in the hydrological and carbon cycles throughout the whole Cenozoic, which is also one of the focuses of this famous paper. Was the 405 kyr cycle not as important as the precession in the late Cretaceous? At least, you should point out its role and clarify its relationship with the precession in the late Cretaceous tropical Pacific Ocean rather than ignore it and made no change in the revision.

The x-axis in figure S12b (MTM spectral analysis) was labeled with the unit cycles/meter but should have been cycles/kyr. This is a typo and we are grateful that the reviewer pointed out this mistake. It has been corrected in the revised supplement.

Section 4.2 "Orbital cyclicity in the tropical Pacific biological pump" describes the relative importance of the various cycles seen in our records of bulk carbon and oxygen isotopic composition and XRF Ba record. As we mentioned in our response previously to the reviewer, "…there is a strong 405-kyr component in both carbon and oxygen bulk isotope data. We are cautious to interpret the oxygen bulk stable isotope data because it will be a mix of calcareous nannofossils from the surface ocean, planktonic foraminifera from the upper ocean layers and, to a lesser extent, the deep sea benthic foraminifera. It is not focus of this manuscript." The CENOGRID record was constructed from deep sea benthic foraminifera – and the 2020 paper was focused on interpreting these global changes. The focus of this manuscript is on the export production record (XRF-Ba) in the tropical Pacific Ocean. We do include initial observations on the bulk record in this section (1st paragraph) "…Because the bulk carbonate $\delta^{13}$C values closely resemble the lower resolution surface planktic foraminiferal $\delta^{13}$C record of *Rugoglobigerina rugosa* (Fig. S3) from the same sites reported by Jung et al. (2013), it is possible that it reflects surface conditions potentially related to local surface productivity. However additional high resolution work is needed to confirm this initial observation." In the Section 1. Introduction (2nd paragraph) we state that "…Proxy records that provide evidence of carbon cycle dynamics, such as marine carbon isotope records ($\delta^{13}$C), show dominant variability in the eccentricity (rather than precession) band. This effect could be due to the long residence time of carbon in Earth's exogenic system, which filters out higher resolution fluctuations (e.g., Pälike et al., 2006) or is related to orbitally paced phytoplankton evolution (Beaufort et al., 2022)."

Based on the spectral analysis shown in Figure S12, the XRF Ba record is clearly dominated by short cycles in the order of 20-30 cm (highest peak in the MTM) and a really minor component of a 5 m cycle, which is equivalent to the long eccentricity cycles if the short cycles are precession (~20 precession cycles in one 405-kyr cycle; ~20 25 cm cycles in a 5 m cycle). The one dominating rhythm for the carbon export record is related to precession with very little expression of modulations by eccentricity. This is in spite of the fact that the age model is based on the 405-kyr dominant cycle in bulk carbon isotopes (~ 5 m cycle). So, yes, the 405 kyr cycle is not as important as precession for carbon export in the tropical Pacific in the late Cretaceous. This is the focus of the manuscript, including the title.

The reviewer points to other cycles that might be of significance writing: "Even though this paper focuses on the XRF-Ba derived productivity record, as they stressed in the rebuttal letter, the most abundant and significant spectral peaks that range from 0.025 to 0.06 (cycles/kyr) include the cycles of both the precession and non-precession bands … Thus, what do these non-precession cycles represent? These non-precession cycles are as significant as the 19-23 kyr precession cycles."

The band 0.025 to 0.06 (cycles/kyr) is the range from 40 to 16.7 kyr thus spanning the range from obliquity to precession. The age model is based on a very simple and minimalistic 405-kyr cycle level preventing the introduction of obliquity and precession components into the spectral analysis allowing a clear view to the cycle distribution of the XRF Barium record. The MTM spectral power analysis (Figure S12a) in the depth domain shows a broader high significance interval of cycles ranging from 50 cm to a little less than 20 cm per cycle. The 405-kyr age model allows to identify those cycles as mainly in the precession frequency band. There is no single sharply defined dominant peak for precession because precession has several components (mainly 19 and 23 kyr, and some others) AND the record is affected by changes in sedimentation rate and drilling disturbance. Thus there will be no clear spectrum. The MTM spectrum is a statistical tool to verify what can be seen by the human eye in the record and it should not be overinterpreted in terms of the significance levels. The algorithms behind it are definitely not made for geological data and thus a robust significance level is problematic.

Therefore, we think that the cycles seen in the power spectra are related to orbital cycles, mainly precession, and are not non-precession related cycles as suggested by the reviewer. We thus refrain from changing the basic assumption as this is backed up by the very good correlation to the Zumaia record.

The reviewer reiterates: "We have no doubt that the precession plays an important role in modulating the XRF-Ba derived productivity changes. However, the spectral and wavelet analyses also tell us that the other orbital cycles including those near the precession band and the 405-kyr long eccentricity cycle also play significant roles. This is the reason why I commented on the original draft that why do you only concentrate on the precession band?"

We think that based on the MTM analysis of the XRF Ba data precession is the major component in the cyclicity as explained above. Eccentricity does play a modulating role but is of much less significance and presence in the data. We do not share the opinion of the reviewer here and are not convinced that precession and eccentricity play an equal role in the XRF Ba data. As provided in the first reply to the reviewer we think that for the productivity changes recorded in the Ba record, not the other records, the data speak for themselves showing clearly a clear imprint of precession cycles that is focus of the manuscript, not the minor component of eccentricity.

In addition, the reviewer suggests: "The CENOGRID climate records from the tropical oceans (Westerhold et al., 2020, Science), a great work led by one of the corresponding authors of this manuscript, display dominant 405-kyr cycles in the hydrological and carbon cycles throughout the whole Cenozoic, which is also one of the focuses of this famous paper. Was the 405 kyr cycle not as important as the precession in the late Cretaceous? At least, you should point out its role and clarify its relationship with the precession in the late Cretaceous tropical Pacific Ocean rather than ignore it and made no change in the revision."

We did not ignore the 405 kyr cycle in our data. We used the imprint in the bulk carbon isotope data to develop a simple 405-kyr eccentricity age model. The strong 405-kyr component in bulk d13C data in the Cretaceous is well known. Prominent examples of eccentricity modulated precession cycles in geochemical data is the late Paleocene and early Eocene (see Lourens et al. 2005; Westerhold et al. 2007 and 2008) are published from Walvis Ridge in the South Atlantic. Compared to those records the Maastrichtian Shatsky Rise XRF records show a minor eccentricity component. Because of the modulation of the precession by eccentricity some related variation in amplitude of the data can be expected and is seen, but not to an extent that would lead to the interpretation of major importance on the XRF data. In the manuscript we focus on the cyclicity of XRF Ba data as a proxy for carbon export which

we show are very likely related to changes in surface ocean productivity. And because these data are dominated by precession our manuscript focuses on the precessional pacing of tropical ocean carbon export. We think this is justified by the outstanding quality of the data. As the manuscript is aiming to explain the dominant precession component and not in addition the globally seen dominant 405-kyr eccentricity, we would refrain from discussing more in the current manuscript this aspect as asked by the reviewer as it will imply a major expansion of the existing manuscript and distract from the focus of this manuscript. As a side note, the benthic foram data for example only cover the potential precession cycles seen in the XRF Ba and do not extend to reconstruct changes in 405-kyr eccentricity. We are trying to understand these dominant cycles of precession in the XRF Ba record.

We highlight the novel aspect of the focus of this manuscript on the cycles of carbon export in the tropical Pacific during the Maastrichtian using the new composite XRF record at Shatsky Rise - without expanding here in this manuscript (beyond what we have written) on the global 405-kyr eccentricity cycle (which is not dominant in this carbon export record from the tropics). We are excited about the opportunity to share these results with others in this manuscript which we hope is publishable in *Climate of the Past.*
* * *
Referee comment – signed by Mingsong Li: I have no further comment. This manuscript can be published as is.

We thank the reviewer (Mingsong Li) for their review and support for publication.